# GROOT: Graph Edge Re-growth and Partitioning for the Verification of Large Designs in Logic Synthesis

## Abstract

Traditional verification methods in chip design are highly time-consuming and computationally demanding, especially for large scale circuits. Graph neural networks (GNNs) have gained popularity as a potential solution to improve verification efficiency. However, there lacks a joint framework that considers all chip design domain knowledge, graph theory, and GPU kernel designs. To address this challenge, we introduce GROOT, an algorithm and system co-design framework that contains chip design domain knowledge and redesigned GPU kernels, to improve verification efficiency. More specifically, we redesign nodes features utilizing the circuit node types and the polarity of the connections between the input edges to nodes in And-Inverter Graphs (AIGs). We utilize a graph partitioning algorithm based on the observation that approximately only 10% boundary edges (nodes) between cluster, to divide the large graphs into smaller sub-graphs for fast GPU processing. We carefully profile the EDA graph workloads and observe the uniqueness of their polarized distribution of high degree (HD) nodes and low degree (LD) nodes. We redesign two GPU kernels (HD-kernel and LD-kernel), to fit the EDA graph learning workload on a single GPU. We evaluate the performance of GROOT on large circuit designs, e.g., Carry Save Adder (CSA) multipliers, the 7nm technology mapped CSA multipliers and Booth Multipliers. We compare the results with state-of-the-art GNN-based GAMORA and the traditional ABC framework. Results show that GROOT achieves a significant reduction in memory footprint (59.38 %), with high accuracy (99.96%) for a very large CSA multiplier, i.e. 1,024 bits with a batch size of 16, which consists of 134,103,040 nodes and 268,140,544 edges. We compare GROOT with state-of-the-art GPU-based GPU Kernel designs such as cuSPARSE, MergePath-SpMM, and GNNAdvisor. We achieve up to $1.104\times$, $5.796\times$, and $1.469\times$ improvement in runtime, respectively.

## 1 Introduction

Logic synthesis plays a vital role in chip design by converting high-level circuit descriptions into optimized gate-level implementations and helps to bridge the gap between high-level synthesis and physical design (7). Verification is a critical step in logic synthesis that ensures internal functionality, prevents costly errors, and reduces the time-to-market by identifying and fixing issues early in the design cycle (19). However, traditional verification methods are computationally demanding and increasingly time-consuming, especially for complex designs (8; 28). For example, as measured in (20), the verification process takes more than 100 hours for the booth multiplier using the OneSpin commercial equivalence checker tool. Furthermore, using the open-source verification tool ABC (15), a 2048-bit multiplier requires $8.6 \times 10^5$ seconds (more than nine days) (25).

Graph neural networks (GNNs) have gained popularity as a potential solution to improving verification efficiency, e.g., GAMORA (25), since graph is one the most natural ways to represent many fundamental objects in circuits, such as Register Transfer Level (RTL) descriptions, netlists, layout, and Boolean functions. In GNN-based methods, GNN is leveraged to classify the graph nodes which significantly reduces the verification time. For example, the 2048-bit multiplier verification time reduces from more than nine days to 0.919 seconds when GNN is used (25).

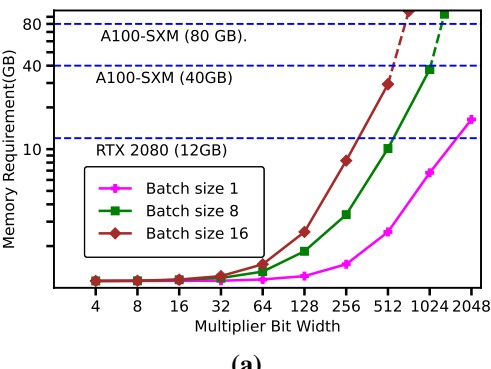

Table 1: Comparison of verification methods.

| Methods | Large multipliers | runtime | accuracy |
|---|:---:|:---:|:---:|
| **ABC (15)** | ✓ | ✗ | ✓ |
| **GAMORA (25)** | ✗ | ✓ | ✓ |
| **GROOT (ours)** | ✓ | ✓ | ✓ |

**(b)**

**(a)**

Figure 1: (a) Extremely high GPU memory requirements on large circuit graphs in EDA. Example: verification of Carry Save Adder (CSA) multiplier with different bits and batch sizes in logic synthesis. (b) Comparison of verification methods.

Despite their promising results, there are research gaps. **First**, an effective graph machine-learning solution for logic synthesis requires a fusion of electronic design automation (EDA) domain expertise and knowledge of graph machine learning. However, existing efforts tend to focus on just one aspect, such as applying GNN algorithms to EDA tasks, and may lack EDA domain expertise. For instance, GAMORA (25) does not distinguish Primary Inputs and Primary Outputs (PO) when creating graph node features, however, PI and PO are inherently different and need to be distinguished. **Second**, processing a large-scale EDA GNN on a single hardware, which is crucial to efficient AI, has been largely neglected. Figure 1 shows the memory consumption (on two high-end GPUs NVIDIA A100 40 GB and 80 GB, and one low-end GPU GeForce RTX2080) required for the verification of various bit widths multipliers. We observe that even the NVIDIA A100 could not accommodate the 1,024-bit CSA multiplier graph when batch size equals 16. Please note that batch processing is essential to achieve high throughput as GPUs are designed to process parallel data. **Third**, the state-of-the-art (SOTA) high-performance solutions often use GPU, and simply adopt commercialized multi-GPU solutions (e.g., GAMORA directly uses Pytorch Geometric (5) on two or more GPUs). However, an important aspect that frequently goes unnoticed is the consideration of GPU primitives. This fundamentally limits making single GPU achievable for EDA GNN and the applicability of broadening accessibility in economically disadvantaged districts.

In this research, we propose GROOT, Graph Edge Re-growth and Partitioning for the Verification of Large Designs in Logic Synthesis. GROOT is a single-GPU-based framework and simultaneously achieves high accuracy, and low memory footprint at run-time. The classical open-source EDA tool ABC (15) is not capable of obtaining verification results at run-time, and GAMORA (25) faces the out-of-memory issue on large circuit graphs, as summarized in Table 1.

Our key contributions are: (i) We create the EDA graph node features. We utilize the circuit node types and the polarity of the connections between the input edges to nodes in And-Inverter Graphs (AIGs), to form the input embedding of the EDA graph. With the addition of more features, our GNN model possesses the capacity to learn from a broader spectrum of circuit characteristics. (ii) At the graph processing level, we utilize a graph partitioning algorithm to divide the large graphs into smaller sub-graphs for GPU processing and develop a boundary edge re-growth algorithm. (iii) We carefully profile the EDA graph workloads and observe the uniqueness of their polarized distribution of high degree (HD) nodes and low degree (LD) nodes. We redesign two GPU kernels (HD-kernel and LD-kernel), to fit the EDA graph learning workload on a single GPU.

## 2 RELATED WORK

**Verification:** Verification can be performed at multiple stages (see Appendix 7.1) to ensure that the designed chip meets its intended functionality. Traditional formal verification techniques include Satisfiability (SAT), canonical diagrams, theorem proving (4), and algebraic re-writing. The SAT technique models the verification problem as Boolean satisfiability (27; 9). Canonical diagrams propose different graph-based representations, such as binary decision diagrams (BDDs) (1), Taylor expansion diagrams (TEDs) (2), and binary moment diagrams (BMDs) (17). The algebraic

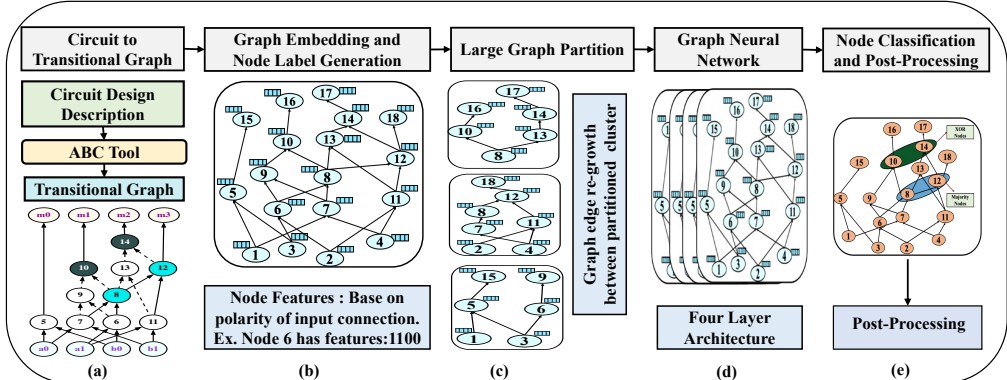

Figure 2: Framework Overview: (a) Circuit to Transitional graph conversion. (b) EDA graph with node features. (c) Large Graph Partitions to solve GPU memory issues. (d) Graph Neural Network architecture. (e) Node classification and post processing.

approaches, based on modeling circuit specifications and hardware implementation as polynomials (18), leverage symbolic computer algebra techniques (12; 3; 9) to solve verification problems.

**GNN in Circuits.** GNNs efficiently learn graph-like structures and extracting information (10), particularly in EDA, where circuit netlists can be naturally represented as graphs. For instance, NeuroSAT (21) employs message passing in a neural network to learn SAT problems and predict satisfiability. In another study (11), GNNs were utilized to predict testability analysis for netlists and demonstrated performance comparable to commercial tools. Further work is required to optimize their performance and overcome scalability and data management limitations, enabling their full effectiveness in circuit verification (25).

## 3 GNN FOR VERIFICATION IN LOGIC SYNTHESIS

The overview of GROOT framework is depicted in Figure 2, consisting of five stages, i.e., (a) Converting the netlist into a transitional graph representation using an open-source EDA tool ABC (15); (b) Pre-process the transitional graph and generate the standardized logic synthesis-based EDA graph; (c) Partition of the large EDA graphs; (d) Utilize GNN for aggregation and message passing; and (e) Node classification and post-processing.

### 3.1 CONVERTING NETLISTS INTO TRANSITIONAL GRAPH

A Boolean network (digital design) can be described as a directed acyclic graph (DAG), where the nodes symbolize logic gates. An And-Inverter Graph (AIG) represents a specific type of combinational Boolean network, comprised of two input `AND` gates and `inverters` (14). Essentially, AIG graphs are specialized DAGs that encapsulate the logical functionality of Boolean networks. Interestingly, through DeMorgan's rule, the combinational logic of any given Boolean network can be easily transformed into an AIG.

In GROOT, this transformation is accomplished through an open-source EDA tool ABC (15). Figure 3 illustrates this process using a two-bit CSA multiplier. The ABC takes a netlist as an input, as shown in Figure 3 (a), and generates the corresponding AIG representation, shown in Figure 3 (b). In AIG representation, inputs $a1a0$ and $b1b0$ represent the two-bit binary numbers for the multiplier and multiplicand, respectively. The multiplication result is represented using $m3m2m1m0$ bits. For example, multiplier $a1a0 = 10$ and multiplicand $b1b0 = 11$ gives the multiplication result $m3m2m1m0 = 0110$. The multiplication of the least significant bits (LSB) highlighted in golden color, symbolizing the 'AND' operation at node 5 (i.e., $m0 = a0 \cdot b0$). The additional operation of multiplication is 'XOR' (green), containing nodes 6, 7, 8, 9, and 10, and can be represented by the equation $m1 = a0 \cdot b1 \ xor \ a1 \cdot b0$. The 'NOT' operations are indicated by dashed lines.

### 3.2 NODE FEATURES AND NODE LABEL CREATION

We take the node and edge information from the AIG representation (transitional graph) to form the EDA graph. We define circuit-based EDA graph as $G = (V, E)$ with $N$ nodes $v_i \in V$ and

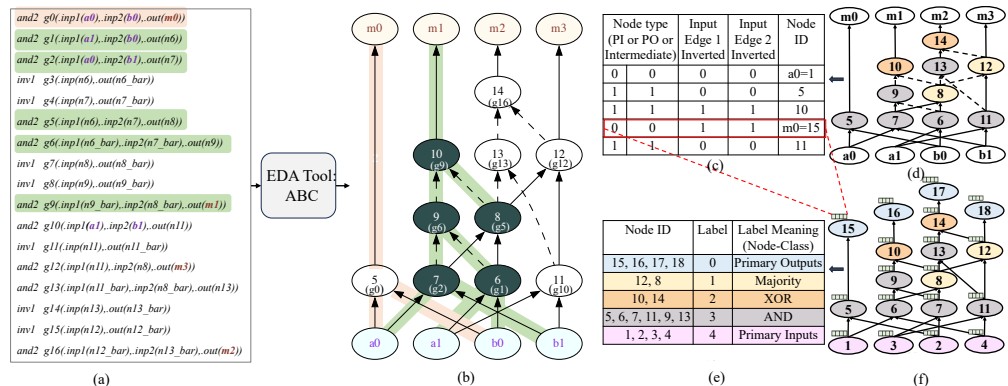

Figure 3: Input to EDA graph flow: (a) Two-bit multiplier netlist. (b) AIG representation of two-bit multiplier using ABC (the dotted line represents inverted inputs to node). (c) Node features (selected nodes shown). (d) EDA graph of two-bit CSA multiplier. (e) Ground truth labels for the GNN model. (f) EDA graph embedding with node features of two-bit CSA multiplier.

edges $(v_i, v_j) \in E$. We use an adjacency matrix $A \in R^{N \times N}$ to describe graph connections, a degree matrix $D_{ii} = \sum_j A_{ij}$ and a feature matrix $X = \{x_1, x_1, ..., x_N\}$. We create the input embedding graph utilizing four distinct node features. The nodes in an EDA graph (Figure 3, (d)) can be categorized into three distinct types: Input variable nodes or Primary Inputs (PI), Logic gate nodes or internal nodes which are AND gates, and Output variable nodes or Primary Outputs (PO). We create node features from the node types and the polarity of input edges as depicted in Figure 3 (c). The first two bits indicate the node, e.g., PI, internal node, or PO. The encoding is as follows: PI and PO are represented by '00'. Internal nodes are represented by '11'. The subsequent two bits are used to characterize the polarity of the input edge connections. For instance, node 5, an internal node with non-inverted input edges, has a feature vector of 1100 as depicted in figure 3 (c). Similarly, node 10, another internal node with inverted inputs, has a feature vector of 1111. The PI node 1 or $a0$ has a feature vector of 0000, while the PO node 15 or $m0$ has a feature vector of 0011 as highlighted in red dotted lines between the figures 3 (c) and 3 (f).

We create EDA graph embedding using these node features as shown in Figure 3 (f). Our input embedding contains four-node features, a distinction from the three-node features in GAMORA (25). Implementing additional node features offers a more robust representation of nodes and improved generalization. Our model possesses the capacity to learn from a broader spectrum of circuit characteristics. Next, we formulate labels for the ground truth using ABC (15). Figure 3 (e), depicts the labels for the two-bit CSA multiplier. For nodes 1 to 4 (PI nodes), we label them as 4. For nodes 5, 6, 7, 9, 11, 13 (two-input AND gates), we label as 3. For nodes 10 and 14 (XOR), we label as 2. For nodes 12 and 8 (MAJ functionality) are labeled as 1. Lastly, all PO nodes, namely 15 to 18, are labeled as 0.

### 3.3 PARTITION, NODE CLASSIFICATION AND POST-PROCESSING

To deal with the memory footprint challenge caused by large EDA graphs, we use the graph partition, where we divide our graph into sub-graphs as shown in Figure 2 (c), and feed them to our GNN architecture to perform a node classification task (see Appendix 7.2). We use the GraphSAGE framework (6), a "sampling-and-aggregation" approach to generate node representations. We observe that EDA graphs contain approximately only 10% boundary edges (nodes) between clusters. We regenerate boundary edges between disconnected clusters to prevent the loss of features and support effective message passing between inter-cluster nodes (see boundary edge re-growth algorithm in Appendix 7.3).

We use GNN to classify the nodes into two categories XOR and MAJ as depicted in Figure 2 (e). We use the algebraic re-writing technique (28; 3) for verification. The algebraic representation of the basic Boolean operators is summarized in the appendix Table 3. Consider the case involving the XOR and MAJ operations. The sub-polynomial expression is $x_1 + 2x_2 + \ldots$, where $x_1 = \text{XOR}(a, b, c)$ and $x_2 = \text{MAJ}(a, b, c)$, where $a, b, c$ are inputs of XOR and MAJ functions. Substituting the algebraic representations of XOR and MAJ into the sub-polynomial, we obtain, $x_1 + 2x_2 + \ldots = (a + b + c - 2ab - 2ac - 2bc + 4abc) + 2(ab + ac + bc - 2abc) = a + b + c$.

This simplification results in the elimination of the four nonlinear terms: $2ab$, $2ac$, $2bc$, and $4abc$. This polynomial reduction based on algebraic re-writing (28; 3) is integrated in ABC (15). This approach is reliant on detecting XOR and MAJ gates from a flattened netlist, a process that tends to be time-consuming. We leverage the GNN node classification to detect XOR and MAJ gates which makes verification efficient. In the two-bit CSA multiplier, nodes 10 and 14 are classified as XOR, while nodes 12 and 8 are classified as MAJ. These nodes are subsequently used for verification with the methodology described in (28).

## 4 KERNEL DESIGN - GROOT-GPU

We tailor GPU kernels (high-degree (HD) kernel and low-degree (LD) kernel) separately for the extremely high-degree macro nodes ($\geq$ 512) and the low-degree macro nodes ($\leq$ 12) (EDA graph observation please see Appendix 7.4). The whole GPU kernel is programmed in CUDA C. The codebase will be released with the paper. We start by partitioning the workload (non-zero elements) statically for each row of the adjacency ($A$) matrix (all nodes possessing a degree equal to the width). This involves splitting the non-zero elements evenly into $2^n$ parts, then sequentially assigning these divisions to distinct warps within the block, repeating until all rows' workload has been allocated.

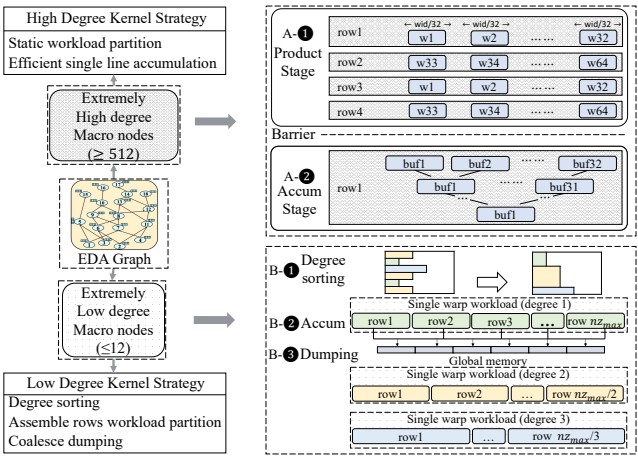

Figure 4: GPU Kernel Design for EDA

We show an example in the Fig. 4. The HD macro nodes contain 4 rows, namely $row1$ to $row4$. Each row contains $wid$ non-zero elements. Each block in the kernel contains 64 warps, numbered from $w1$ to $w64$. We divide each row into 32 equal workloads, each containing $\frac{wid}{32}$ non-zero elements. Then we assign the workloads in $row1$ to the warps numbered $w1$ to $w32$ in turn, and assign the workloads in $row2$ to warps from $w33$ to $w64$. Repeat the above process, and assign the workloads in $row3$ and $row4$ to warps from $w1$ to $w32$, and warps from $w33$ to $w64$ in turn. In step A-1, the kernel multiplies non-zero elements in the adjacent matrix with the corresponding rows of the right-hand feature matrix based on static workload partitioning. Intermediate results are stored in shared memory buffers assigned to each warp. Each warp, from $w1$ to $w32$, has its corresponding buffer ($buf1$ to $buf32$). In step A-2, we accumulate the results in the 32 buffers using the tree-based accumulation. First, the direction of the warp operation is reset, and the 32 threads in each warp are responsible for one bit in the buffer with the same number. Then, the tree-shaped accumulation operation within the warp is performed using warp synchronization primitives, which can be completed in 5 cycles. The tree-shaped accumulation, designed to ensure efficient completion, can save about half the number of cycles compared to the AtomicAdd function to complete all accumulations. As shown in the example in figure 4, after the first cycle, we get the first accumulation result, that is, the sum of $buf1$ and $buf2$ is stored in $buf1$, the sum of $buf3$ and $buf4$ is stored in $buf3$, ..., and the sum of $buf31$ and $buf32$ is stored in $buf31$. Following the same process, after the second cycle, we get the second accumulation result stored in $buf1$, $buf5$, ..., $buf29$. After the fifth cycle, the final accumulation result is stored in $buf1$. After the tree-shaped accumulation operation is completed, the final output of a row of results is obtained, which can be directly transferred to the corresponding row in the global memory without additional accumulation operations on the global memory, greatly increasing the execution efficiency of this SPMM kernel.

The LD-kernel design for low-degree macro nodes is shown in the lower half of Fig. 4. Step B processes the degree sorting on the adjacent matrix with the following steps: (1) computing each row's degree using the row pointer array with time complexity of $\mathcal{O}(n)$ when employing count sort (23) or radix sort (13), with $n$ indicating the number of rows; (2) applies a stable sorting algorithm to sort rows by their degrees; and (3) updating the row pointer array to reflect the new rows' order, with

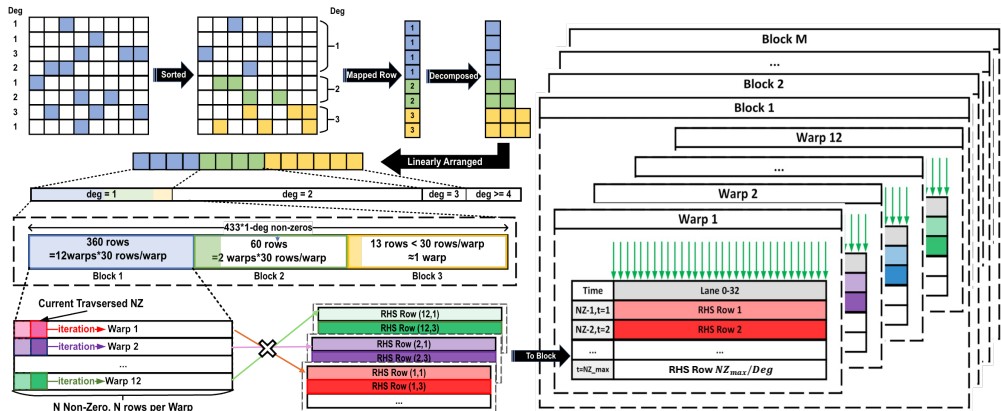

Figure 5: Detailed process of LD-kernel, from degree-sorting, row-assembling, block-partitioning, to warp-wise multiplication and summation, with block-wise parallelism.

time complexity of $\mathcal{O}(n)$. The dominant time complexity of this operation arises from applying the stable sorting algorithm. Nevertheless, employing count sort, a linear time-complexity algorithm, can optimize the overall time complexity to $\mathcal{O}(n)$. This enhances efficiency compared to alternative algorithms and the rearranged adjacent matrix has a highly regular degree distribution due to partitioning. We adopt the "row-assembling" workload partitioning in step B-2, which is different from GNNAdvisor (24). This approach assigns multiple rows at the same degree to one warp to achieve a higher rate of utility of a single warp, thereby increasing the overall efficiency. We set this kernel's number of warps per block to $warp_{max}$ as a hyperparameter. In the example, for rows with degrees of 1, 2, and 3, each warp is responsible for $nz_{max}$, $nz_{max}/2$, and $nz_{max}/3$, rows, respectively, where $nz_{max}$ is also a hyperparameter indicating the maximum amount of non-zeros each warp can contain. When the degree is one, $row_1$, to $row_{nz_{max}}$, are assigned to the workload of a single warp. Similarly, $nz_{max}/2$, rows at the degree of 2 and $nz_{max}/3$, rows with a degree of 3 are assigned to two separate warps. In addition, the partitioning remains is processed recursively to ensure the minimum error in calculation, as mentioned in part 2. The whole partitioning method significantly improves the efficiency of the LD-kernel.

The sorting and row-assembling details are described in Fig. 5. The warp-block operation of LD-kernel starts with sorting on the original sparse input by the degree of each row and maps the left-hand side (LHS) rows into an array linearly, whereby the partitioning is executed via dividing the array into blocks of rows according to their degree, sequentially from the smallest to the largest. Then, within the block, warps will operate in parallel to extract non-zeros in the rows to multiply the corresponding right-hand-side (RHS) rows. The resulting product rows are summed up to produce the output. All blocks perform the whole process in parallel. For example, in the lower left part of Fig. 5, warp 1 traverses its non-zeros from the left to the right, meanwhile locating the corresponding RHS row (1,1). The first 1 implies warp 1, and the second 1 refers to the first row, which warp 1 is responsible for.

Multiple rows of non-zero elements are assigned to the same warp rather than one row per warp. Since the degree of each row is small, say 3, the number of warps per block in this kernel is set to $6m$. For rows with degrees of 1, 2, and 3, each warp is responsible for $6m$, $3m$, and $2m$ rows, which translates into $nz_{max}$, $nz_{max}/2$ and $nz_{max}/3$ rows in the figure, respectively. This increases the proportion of active warps. In addition, this significantly reduces the overhead of warp switching and improves calculation efficiency. The aggregated organization method enhances access continuity to global memory when transferring calculation results from shared memory to global memory. On top of that, We utilize the technique of coalesce dumping to exploit this advantage fully. During step $B-3$ called coalesce dumping, it writes the intermediate results of multiple consecutive rows from the same warp into a continuous area in global memory, its efficiency is also improved.

## 5 PERFORMANCE EVALUATION

This section presents the evaluation of GROOT by comparing its memory usage, and run-time against two baselines: the traditional open source tool ABC (15) and the state-of-the-art GNN-

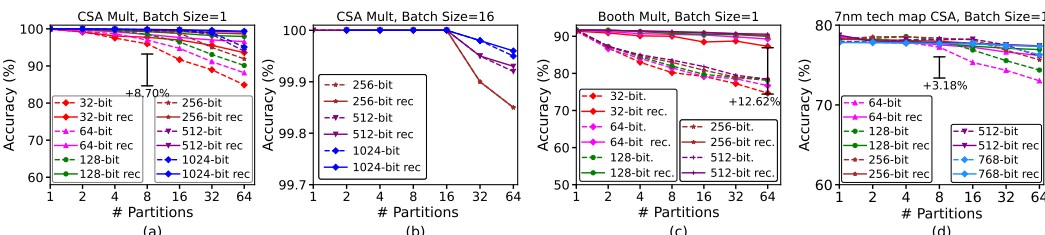

Figure 6: Verification accuracy as a function number of partitions for (a, b) CSA multipliers, batch size 1 and 16, respectively; (c) Booth multipliers, batch size 1; (d) CSA multipliers after 7nm technology mapping, batch size 1. All the multipliers were trained using 8-bits.

based GAMORA (25). When comparing with (25), we run the GAMORA framework against our modified datasets. We use a Linux-based host with AMD EPYC 7543 32-Core Processor and an NVIDIA A100-SXM 80 GB.

**Dataset Generation.** *Carry Save Adder (CSA) Multipliers.* We create a dataset of CSA multipliers utilizing the open-source tool ABC (15; 28). We partition the dataset into splits of 80% for training, 10% for validation, and 10% for testing. We generate input graph embeddings with features and labels essential for GNN-based learning. A noticeable trend is the significant increase in the number of nodes and edges of the input embedding graph as the bit widths of CSA multipliers increase. For instance, multiplier of 1024-bit width has around 8.3 million nodes and 16.7 million edges (Appendix Table 4). Furthermore, we create a dataset of large CSA multipliers to evaluate GROOT on larger graphs. We set the batch size to 16 and created a very large graph (Appendix Table 5).

*Booth Multipliers.* We create a new dataset dedicated to Booth multipliers to expand the dataset options (See Appendix Table 6) Booth-encoded multipliers, compared to CSA multipliers have complex structures and produce more complex graphs.

*CSA Multipliers after Technology Mapping.* For a thorough evaluation of GROOT, we use the ASAP 7nm technology mapping (26) on CSA multipliers to create a 7nm technology-mapped dataset. This integration with the technology-mapped netlist offers an additional dataset intended for post-technology mapping (See Appendix Table 7). Further, we create an FPGA-mapped CSA multiplier dataset to evaluate GROOT.

### 5.1 ACCURACY ANALYSIS OF VARIOUS MULTIPLIERS

Our GNN model is trained on an 8-bit multiplier and then used in inference on larger multipliers of the same dataset. For instance, the model is trained using an 8-bit CSA multiplier and is then tested on a 1024-bit multiplier, with a batch size of 16. Figure 6 shows the accuracy across various datasets as a function of the number of partitions. In all figures, solid lines indicate accuracy with recovery, and dashed lines represent accuracy without recovery. Figure 6 (a) shows the accuracy on CSA multipliers with batch size one. Without any partitioning (i.e., number of partition=1), we achieve high accuracy, reaching 100% for multipliers of sizes 128-bits and above, while the accuracy is 99.94% for the 32-bit multiplier. As the number of partitions increases, the loss in accuracy becomes more noticeable because more partitions require the removal of more boundary edges. However, using our boundary edge re-growth approach effectively recovers accuracy. In Figure 6 (a), the solid line denotes the regained accuracy when using boundary edge re-growth. Notably, this edge re-growth method achieves a maximum recovery of 8.7% boost in accuracy of a 32-bit multiplier. By adopting our edge re-growth approach, one can afford to use more partitions while maintaining high accuracy.

We evaluate accuracy on large CSA multipliers such as the 1024-bit multiplier with a batch size of 16 containing 134,103,040 nodes and 268,140,544 edges. The figure 6 (b) shows accuracy with respect to number of partitions. The trends show that the accuracy is at 100% up until 16 partitions. This accuracy can be attributed to the presence of a large number of edges in these large graphs and a small number of edges removal does not affect message passing in GNN. Consequently, partitioning does not much impact the accuracy. However, post the 16-partition mark, there is a slight drop in accuracy since more edges are removed to create partitions.

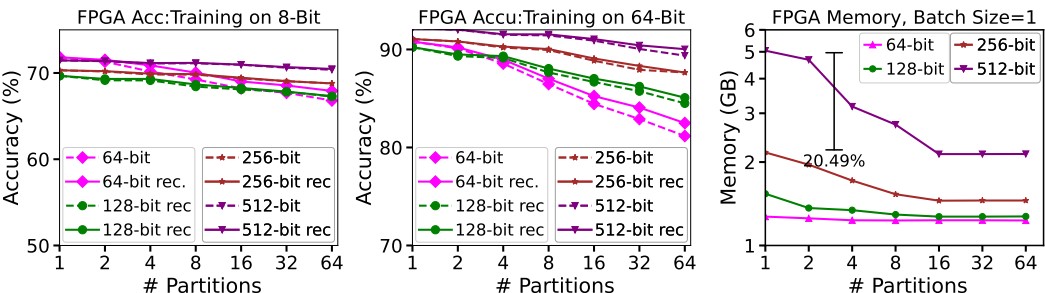

Figure 7: FPGA mapped dataset results showing (a) memory utilization and (b) accuracy as a function of the number of partitions for CSA multipliers, following the application of FPGA mapping, with a batch size of 1. All the multipliers were trained using 8-bits.

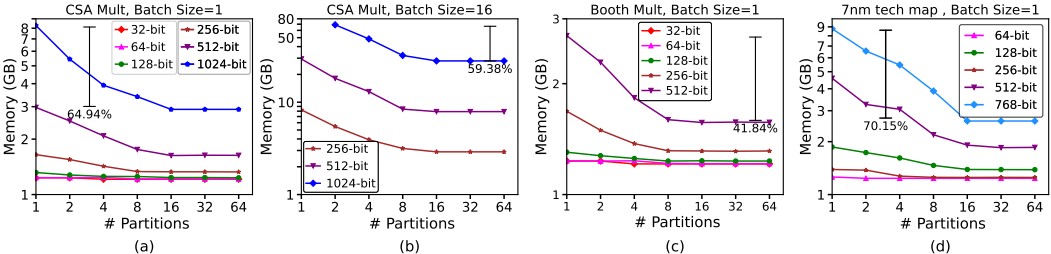

Figure 8: Memory utilization as a function of the number of partitions for (a) CSA multipliers, for a batch size of 1, (b) CSA multipliers, for a batch size of 16, (c) Booth multipliers, for a batch size of 1, and (d) CSA multipliers, following the application of 7nm technology mapping, with a batch size of 1. All the multipliers were trained using 8-bits.

To evaluate GROOT's performance with complex graphs, we utilize the Booth multipliers dataset. Figure 6 (c) shows accuracy with respect to the number of partitions. The accuracy drop is more compared to that in other datasets. However, the utilization of the edge re-growth approach enables the mitigation of this accuracy drop, as illustrated by the solid line in Figure 6 (c). The re-growth achieves a maximum 12.62% accuracy recovery in a 32-bit multiplier. Additionally, we test with the ASAP 7nm technology (26) mapped netlist dataset. This netlist comprises 161 standard cell gates, including the multi-output gate, leading to certain irregularities. As evidenced in Figure 6 (d), even with such irregularity, GROOT shows high accuracy and maintains more than 76% accuracy after edges re-growth. In summation, GROOT is capable of handling design complexities.

Figure 7 (a) shows the accuracy of FPGA-mapped CSA multipliers with batch size equal to 1 and the model is trained on 8 bits. The accuracy is low among all the datasets. To further improve prediction accuracy, we focus on training the GNN model using larger multipliers. This approach significantly improves the model's prediction accuracy. When the model trained on a 64-bit multiplier boosts the accuracy for a 64-bit multiplier from 71.82% (Figure 7 (a), number of partition=1) to 90.8% (Figure 7 (b), number of partition=1) an 18.98% boost in accuracy. However, this accuracy gain comes with increased training time. Training a 64-bit FPGA for 100 epochs takes 2914.42 seconds. To mitigate this time cost, we propose designing a specialized kernel for faster matrix multiplication, a major factor in training and inference time.

## 5.2 MEMORY FOOTPRINT ANALYSIS

Figure 8 illustrates the GPU memory utilization by various multipliers with respect to the number of partitions. Figure 8 (a) shows the GPU memory utilization on the y-axis and the number of partitions on the x-axis for CSA multipliers with batch size one. As the number of partitions increases, the memory requirement decreases. For larger multipliers (e.g., 1024 bits), the memory reduction trend follows an exponential decay. When partitioned into 64 sub-graphs, the 1,024-bit multiplier showed a maximum benefit of 64.94% reduction in memory requirement as per depicted in the Figure 8 (a).

To evaluate the scalability of GROOT, we evaluate its performance on massive multiplier graphs such as the 1024-bit multiplier with a batch size of 16 containing 134,103,040 nodes and

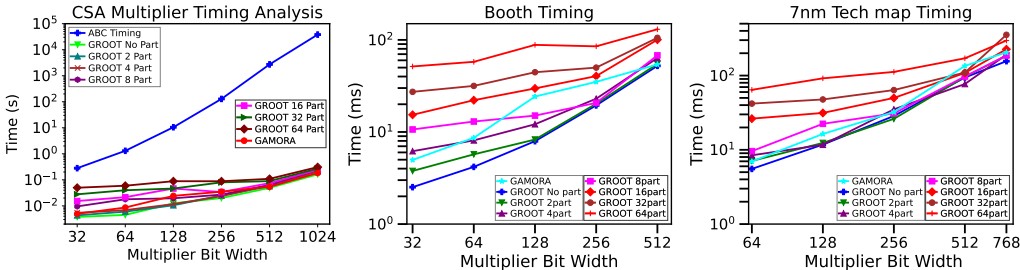

Figure 9: Different multipliers verification time comparisons: (a) CSA Multiplier, (b) Booth Multiplier, (c) 7nm technology mapped multiplier.

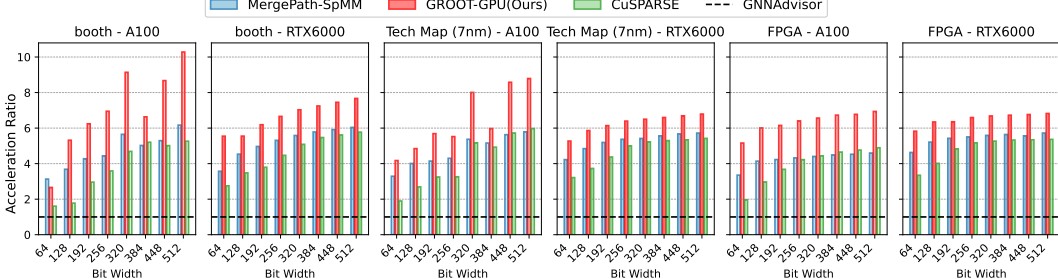

Figure 10: The runtime comparison among GROOT-GPU and SOTA GPU-based GPU Kernel designs, where the acceleration ratio of 1 from GNNAdvisor is drawn as the black dash line.

268,140,544 edges, as depicted in Figure 8 (b). Partitioning the 1,024-bit multiplier into 64 subgraphs resulted in a maximum memory reduction of 59.38%. Without partitioning, even high-end GPUs such as NVIDIA A100-SXM with 80 GB memory cannot perform verification on this massive graph. Thus, GROOT offers a fundamental solution to scalability. Our method is different from GAMORA (25), which requires multiple GPUs to handle massive graphs while we only need one low-end GPU. Table 2 shows the different multipliers and their GPU memory utilization. Furthermore, to recover the accuracy, our algorithm regrows the edges after partitioning. The effect of the number of partitions on the memory requirement can be observed until the number of partitions is equal to 16. When the partitioning size is large (say 32) as shown in Figure 8 (b), the recovered edge consumes a large portion of the memory footprint, thus we observe less memory saving.

To demonstrate GROOT's effectiveness on complex designs, we evaluate it on different complex datasets. Figure 8 (c) illustrates memory utilization versus the number of partitions for booth multipliers, indicating an exponential reduction in memory with respect to partitions. The 512-bit booth multiplier shows maximum memory requirement reduction which is 41.84%. Figure 8 (d), displays memory utilization versus number partitions for 7nm mapped CSA multipliers, demonstrating a significant reduction in memory requirement for post-technology-mapped CSA multipliers. For instance, the maximum memory requirement reduction for the 768-bit multiplier is 70.15%. Similarly, Figure 7 (c) shows memory utilization for FPGA-mapped CSA multipliers. The maximum memory requirement reduction is 57.62% for a 512-bit multiplier. The benefits of memory reduction remain with increased design complexity.

Table 2: Large Multiplier GPU Memory Usage (In MB) Comparison. (Batch size of multipliers=16, OOM= Out of Memory).

| # Part. | 256-Bit | 512-Bit | 1024-Bit |
|---|---|---|---|
| GAMORA | 8,263 | 29,375 | **OOM** |
| GROOT 2 Part. | 5,457 | 18,135 | 68,923 |
| GROOT 4 Part. | 3,923 | 13,025 | 48,463 |
| GROOT 8 Part. | 3,157 | 8,421 | 32,093 |
| GROOT 16 Part. | 2,901 | 7,909 | 27,997 |
| GROOT 32 Part. | 2,901 | 7,909 | 27,997 |
| GROOT 64 Part. | 2,901 | 7,909 | 27,997 |

### 5.3 Run Time Analysis and Comparative Study

Figure 9 shows the inference run time of verifying different multipliers of different widths after applying boundary edge re-growth for accuracy recovery. As shown in Figure 9 (a), as the bit width increases, ABC's run time expands exponentially compared to both GROOT and GAMORA. In comparison, GROOT significantly outperforms ABC (15). When processing graphs for 1,024-bit CSA multipliers partitioned into 64 subgraphs, GROOT achieves a speedup of $1.23 \times 10^5$ over ABC. Furthermore, the verification times exhibited by different partitioned graphs using GROOT align closely with GAMORA (25). It is important to recognize that the verification time for GROOT depends upon the number of partitions: an increase in the number of partitions slightly increases the verification time due to a small partition time. It is important to highlight that neither GAMORA nor ABC can efficiently handle large graph datasets on a single GPU, as depicted in Figure 1.

Further, we show the GROOT verification time using Booth (Figure 9 (b)) and 7nm technology-mapped (Figure 9 (c)) datasets compared to GAMORA (25). We set up the baseline for both these datasets and compared results with GNN-based baseline GAMORA (25) since our earlier result on CSA shows ABC (15) require excessively long verification times. In the case of booth multipliers (Figure 9 (b)), GROOT without partition and with partition equal to two outperforms the verification time of the GAMORA (25). Technology mapped multiplier case (Figure 9 (c)), GROOT with no partition outperforms verification time to GAMORA (25). In both cases, partitioning does not greatly affect runtime, but it improves memory efficiency for large graphs.

### 5.4 GPU kernel results

We compare our design with SOTA GPU-based GPU Kernel designs such as cuSPARSE (16), MergePath-SpMM (22), GNNAdvisor (24). Figure 10 shows the comparison results of MergePath-SpMM, our GROOT-GPU, and CuSPARSE against GNNAdvisor, represented by the black horizontal dashed line. The kernels are tested on the graph of the Booth Multiplier, Technology Mapping, and FPGA 4LUT datasets with bit widths ranging from 64 to 512 and an embedding dimension of 32. The kernels perform SpMM operations given the graph adjacency matrices with corresponding embeddings, and the runtime of SpMM operations are recorded by the type of kernels, the bit width of the net list which graphs describe, and the datasets where the graphs belong to. Our GROOT-GPU demonstrates superior acceleration compared to the other three SOTA SpMM kernels in most cases. The performance gap widens as the bit width of the multiplier datasets increases and with more powerful GPUs. GROOT-GPU achieves the highest acceleration ratio of 10.28 for the Booth dataset with a bit width of 512 on the A100 GPU, outperforming the second-fastest MergePath-SpMM by $1.67\times$ and the third-fastest CuSPARSE by $1.95\times$. The results highlight the efficiency of our GROOT-GPU kernel in SpMM operations, which is an essential step in GNNs' message passing, particularly for complex datasets and higher bit widths, making it a promising choice for various GNN-related applications.

## 6 Conclusion

In this paper, we introduce GROOT, an algorithm and system co-design framework that contains chip design domain knowledge, graph theory, and redesigned GPU kernels, to improve verification efficiency. We redesign nodes features utilizing the circuit node types and the polarity of the connections between the input edges to nodes in And-Inverter Graphs (AIGs). We utilize a graph partitioning algorithm to divide the large graphs into smaller sub-graphs for fast GPU processing. After profiling EDA graph workloads, we notice their distinct distribution of high-degree and low-degree nodes and tailor the GPU kernel accordingly. We evaluate our framework on large circuit designs, e.g., CSA multipliers, the 7nm technology mapped CSA multipliers and Booth Multipliers. We compare the results with state-of-the-arts, e.g., GAMORA and ABC. Experimental results show that GROOT achieves a significant reduction in memory footprint, with high accuracy for a very large CSA multiplier, i.e., 1,024 bits with a batch size of 16. We also compare GROOT with SOTA GPU-based GPU Kernel designs such as cuSPARSE, MergePath-SpMM, and GNNAdvisor, and achieve notable runtime improvement.

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

## 7 APPENDIX

### 7.1 MODERN CHIP DESIGN

In modern chip design (Figure 11), we can perform verification at multiple stages. In this work, we focus on two stages, one is before technology mapping, where we use the CSA multiplier and Booth multiplier as examples. The second one is post-technology mapping, where we use a 7nm mapped CSA multiplier as an example.

### 7.2 GNN AND GRAPH PARTITION

Generally, The EDA graphs are becoming larger due to the scaling of the netlist. by adding more complexity. To use these large EDA graphs for the training or interference in GNN we need large memory GPUs. To deal with the memory footprint challenge caused by large EDA graphs, we use the graph partition, where we divide our graph into sub-graphs as shown in Figure 2 (c), and feed them to our GNN architecture to perform a node classification task. We use the GraphSAGE framework (6), a "sampling-and-aggregation" approach to generate node representations. It randomly samples a small number of neighboring nodes for each node and then uses an "aggregator" neural

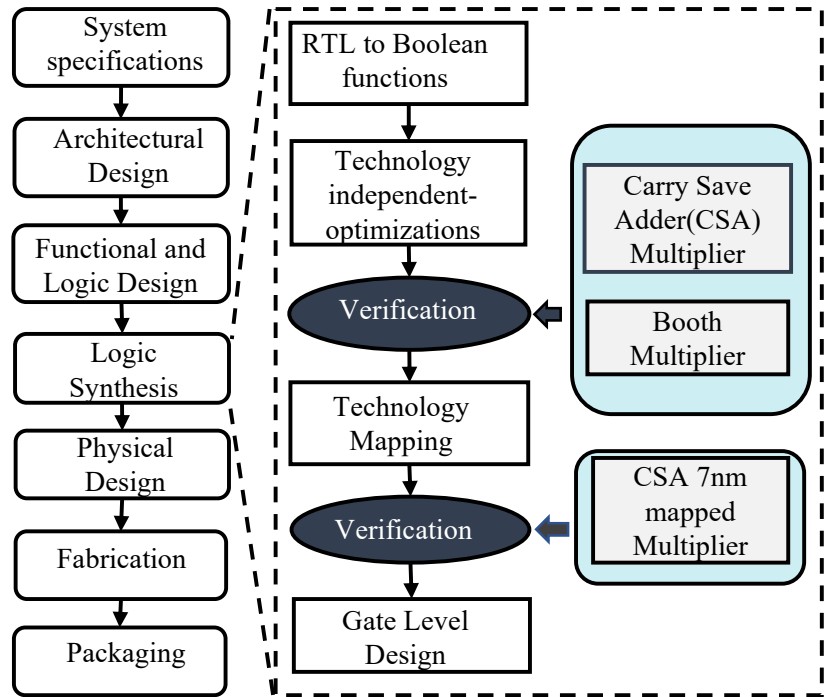

Figure 11: Modern chip design flow and stages of logic synthesis (Colored verification part is our focus and performed by using different designs).

network to combine the representations of the sampled nodes to create a new representation for the original node. This process is repeated multiple times to create a hierarchical representation of the graph. For the given GraphSage model with $K$ layers, the graph embedding propagates between the different layers as follows:

$$\mathbf{h}^k_{\mathcal{N}(v)} \leftarrow \text{AGGREGATE}_k(\{\mathbf{h}^{k-1}_u, \forall u \in \mathcal{N}(v)\}) \tag{1}$$

$$\mathbf{h}^k_v \leftarrow \sigma(\mathbf{W}^k \cdot \text{CONCAT}(\mathbf{h}^{k-1}_v, \mathbf{h}^k_{\mathcal{N}(v)})) \tag{2}$$

In above equations, $\mathbf{W}^k$ denotes weight matrices, $k \in \{1, ..., K\}$; $\sigma$ represents non linear activation; $\mathcal{N}(v)$ denotes the immediate neighborhood function; $\text{AGGREGATE}_k(k \in \{1, ..., K\})$ means the differentiable aggregator function.

This framework is suitable for our EDA circuit graphs because it is designed to work on large datasets. The GraphSAGE layer takes a graph with node features as input shown in Figure 2 (d), and performs a graph convolution operation to aggregate

the node features. This allows the GraphSAGE model to capture and learn the relationships and patterns in the graph data. Multiple GCN layers are stacked in our GraphSAGE model to improve the accuracy and expressiveness of the learned representations.

Table 3: Algebraic Representations of Basic Boolean Operators ( $a, b, c$ are inputs)

| Operation | Boolean Model | Algebraic Model |
|---|---|---|
| NOT | $\neg a$ | $1 - a$ |
| AND | $a \wedge b$ | $ab$ |
| XOR | $a \oplus b$ | $a + b - 2ab$ |
| XOR3 | $a \oplus b \oplus c$ | $a + b + c - 2ab - 2ac - 2bc + 4abc$ |
| MAJ | $(a \vee b) \wedge (a \vee c)$ | $ab + ac + bc - 2abc$ |

### 7.3 BOUNDARY EDGE RE-GROWTH ALGORITHM

Our partitioning algorithm, shown in Algorithm 1, divides a large EDA graph into smaller sub-clusters and facilitates the reconnection of edges between these clusters. We observe that EDA graphs contain approximately only 10% boundary edges (nodes) between clusters, and the boundary recovering process does not add complexity to the inference stage. This approach focuses on regenerating boundary edges between disconnected clusters to prevent the loss of features and support effective message passing between inter-cluster nodes.

---

**Algorithm 1** Graph Partition with Boundary Recovery

---

**Require:** $G, \mathcal{V}$ {Input graph as adj. list and embedding}
1: $[G_0, G_1, ..., G_n], [\mathcal{V}_0, \mathcal{V}_1, ..., \mathcal{V}_n] \leftarrow \text{METIS}(C)$
2: **for** $p = 0$ to $n$ **do**
3: $\quad C_p, N_p \leftarrow \text{FIND\_BOUNDARY\_CONNECTIONS}(G_p)$ {Locate all boundary edges/nodes of partition $G_p$}
4: $\quad G_p \leftarrow G_p \cup C_p$ {Restore the boundary edges}
5: $\quad \mathcal{V}_p \leftarrow \mathcal{V}_p \cup N_p$ {Restore the boundary nodes}
6: **end for**
7: **return** $G_0, G_1, ..., G_n$ and $\mathcal{V}_0, \mathcal{V}_1, ..., \mathcal{V}_n$

---

### 7.4 EDA GRAPH AND KERNEL DESIGN

**Observation.** We analyze some EDA graphs and yield interesting and unique findings. The nodes are split into two categories: one group of nodes with a significantly low degree, e.g. 3 or less for 1024-bit CSA multiplier Figure (Figure 12 (a)), 6 or less for 512-bit 7nm Technology mapped (Figure 12 (b)), 4 or less for 512-bit Booth Multiplier (Figure 12 (c)), and 12 or less in 512-bit FPGA mapped multiplier (Figure 12 (d)); the other group of nodes with significantly higher degrees, e.g., 1024 as shown in Figure 12 (a).

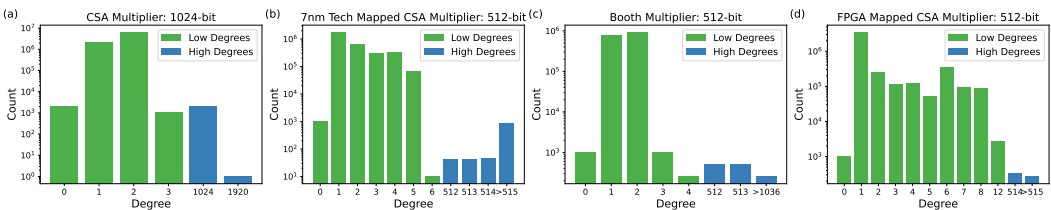

Figure 12: Histogram of Degree distribution of various datasets: (a) CSA multiplier: 1024-bit. (b) 7nm Technology mapped: 512-bit. (c) Booth Multiplier: 512-bit. (d) FPGA mapped multiplier: 512-bit

### 7.5 DATASET STATSTICS

.

*Carry Save Adder (CSA) Multipliers.* Table 4 presents details including the number of nodes, edges, average node degree, and density of each adjacency matrix (with a batch size of 1). A noticeable trend is the significant increase in the number of nodes and edges of the input embedding graph as the bit widths of CSA multipliers expand. For instance, multiplier of 1024-bit width has around 8.3 million nodes and 16.7 million edges. Furthermore, we create a dataset of CSA multipliers to evaluate GROOT on larger graphs. We set the batch size to 16 and generate input graph embeddings with varying bit widths, shown in Table 5.

*Booth Multipliers.* Table 6, displays the statistics of the booth EDA graph.

*CSA Multipliers after Technology Mapping.* Table 7 provides the statistics for graphs derived from the CSA multipliers-mapped netlist.

Table 4: CSA multiplier Dataset Statistics (batch size=1).

| Bit size | # nodes | # edges | Average Degree | Density of A |
|---|---|---|---|---|
| 32 | 7,968 | 21,902 | 2.74 | $3.44 \times 10^{-4}$ |
| 64 | 32,320 | 64,384 | 1.99 | $6.16 \times 10^{-5}$ |
| 128 | 130,176 | 259,840 | 1.99 | $1.53 \times 10^{-5}$ |
| 256 | 1,043,968 | 522,496 | 1.99 | $3.84 \times 10^{-6}$ |
| 512 | 2,093,568 | 4,185,088 | 1.99 | $9.54 \times 10^{-7}$ |
| 1,024 | 8,381,440 | 16,758,784 | 1.99 | $2.38 \times 10^{-7}$ |

Table 5: CSA multiplier Dataset Statistics (batch size=16).

| Bit size | # nodes | # edges | Average Degree | Density of A |
|---|---|---|---|---|
| 256 | 8,359,936 | 16,703,488 | 1.99 | $2.39 \times 10^{-7}$ |
| 512 | 33,497,088 | 66,961,408 | 1.99 | $5.96 \times 10^{-8}$ |
| 1,024 | 134,103,040 | 268,140,544 | 1.99 | $1.49 \times 10^{-8}$ |

Table 6: Booth Multiplier Dataset Statistics.

| Bit size | # nodes | # edges | Average Degree | Density of A |
|---|---|---|---|---|
| 32 | 7,260 | 14,392 | 1.98 | $2.73 \times 10^{-4}$ |
| 64 | 27,852 | 55,448 | 1.99 | $7.14 \times 10^{-5}$ |
| 128 | 108,972 | 217,432 | 1.99 | $1.83 \times 10^{-5}$ |
| 256 | 430,956 | 860,888 | 1.99 | $3.3 \times 10^{-5}$ |
| 512 | 1,713,900 | 3,425,752 | 1.99 | $1.16 \times 10^{-6}$ |

Table 7: Technology Mapped Dataset Statistics.

| Bit size | # nodes | # edges | Average Degree | Density of A |
|---|---|---|---|---|
| 64 | 48,088 | 95,920 | 1.99 | $4.14 \times 10^{-5}$ |
| 128 | 192,487 | 384,462 | 1.99 | $1.03 \times 10^{-5}$ |
| 256 | 769,337 | 1,537,650 | 1.99 | $2.59 \times 10^{-6}$ |
| 512 | 3,084,427 | 6,166,806 | 1.99 | $6.4 \times 10^{-7}$ |
| 768 | 6,949,193 | 13,895,314 | 1.99 | $2.87 \times 10^{-7}$ |

