# OpenReview forum: "GROOT: Graph Edge Re-growth and Partitioning for the Verification of Large Designs in Logic Synthesis"
_ICLR.cc/2025/Conference — ICLR 2025 Conference Withdrawn Submission_

### Official Review · Reviewer_7PxC · 2024-11-01

**Soundness:** 2
**Presentation:** 2
**Contribution:** 2
**Rating:** 5
**Confidence:** 4

**Summary:**

The paper introduces GROOT, a novel framework that utilizes graph partitioning and customized GPU kernel designs to enhance verification efficiency for large-scale circuit graphs. GROOT defines node features based on node type and connectivity, partitions large graphs into sub-graphs for efficient processing, and incorporates two specialized GPU kernels—designed based on node degree—to accelerate training. Experimental results show that GROOT achieves state-of-the-art performance in terms of memory and runtime efficiency.

**Strengths:**

1. The proposed method successfully handles very large circuits with 134 million nodes and 268 million edges, achieving a high accuracy of 99.96%.
2. The novel GPU kernel designs leverage node degree properties, with the HD-Kernel for high-degree nodes and the LD-Kernel for low-degree nodes, delivering state-of-the-art runtime performance.

**Weaknesses:**

1. The related work section could be expanded for a more comprehensive comparison. The paper focuses primarily on comparing with GAMORA as the state-of-the-art, but there are numerous other GNNs in the EDA domain, such as HOGA [1] and DeepGate2 [2]. Additionally, the comparison only considers memory and runtime metrics with GAMORA, whereas it would be beneficial to also compare accuracy with other methods.
2. The paper lacks an ablation study, which is crucial for understanding the contribution of different components. Although the proposed method achieves 100% accuracy on 128-bit multipliers, there is no clear analysis regarding the performance improvement. An in-depth ablation study would help clarify these aspects.
3. The training process is not adequately explained. In Figure 2(c), the large graph is partitioned into sub-graphs, but in Figure 2(d), it appears that GraphSAGE is applied to the entire graph. Including more details about the training process would improve clarity, especially given the paper's emphasis on memory efficiency.

[1] Deng C, Yue Z, Yu C, et al. Less is More: Hop-Wise Graph Attention for Scalable and Generalizable Learning on Circuits. arXiv preprint arXiv:2403.01317, 2024.
[2] Shi Z, Pan H, Khan S, et al. Deepgate2: Functionality-aware circuit representation learning. 2023 IEEE/ACM International Conference on Computer Aided Design (ICCAD).

**Questions:**

1. The paper should include a detailed analysis on why it can achieve such high accuracy.
2. Provide more details about the training process, especially since it directly impacts memory consumption. Including the computational complexity of GROOT compared to other methods would also help readers better understand the memory consumption.
3. Include a broader comparison with more methods, and evaluate performance in terms of accuracy as well.

---

### Official Review · Reviewer_rVyP · 2024-11-03

**Soundness:** 2
**Presentation:** 2
**Contribution:** 2
**Rating:** 3
**Confidence:** 4

**Summary:**

This paper presents GROOT, a Graph Edge Re-growth and Partitioning for the Verification of Large Designs in Logic Synthesis. This GROOT framework consists of five stages, i.e., (1) Convert the netlist into a transitional graph representation using an open-source EDA tool ABC; (2) Pre-process the transitional graph and generate the standardized logic synthesis-based EDA graph; (3) Partition of the large EDA graphs; (4) Utilize GNN for aggregation and message passing; and (5) Node classification and post-processing.

**Strengths:**

+ Logic synthesis is critical for chip design by converting high-level circuit descriptions into optimized gate-level implementations
+ A benchmark is created and numerous experimental simulations were run to showcase the scalability of the GROOT

**Weaknesses:**

- Little information if any is provided behind the reasons of the proposed approach and how it is inspired or improves algorithmically over existing approaches.
- Ablation studies, analysis of the results and implications are incomplete or need a comprehensive restructuring.

**Questions:**

1) The explanation of the reasons behind the partitioning strategy in Section 4 where the paper states “We start by partitioning the workload (non-zero elements) statically for each row of the adjacency (A) matrix (all nodes possessing a degree equal to the width). This involves splitting the non-zero elements evenly into 2^n parts, then sequentially assigning these divisions to distinct warps within the block, repeating until all rows’ workload has been allocated.” are missing or at least unclear. For example, why existing graph partitioning algorithms are not applicable? If the splitting is done as described in section 4, then is the graph representation a useful data structure? Or is it adopted just for the downstream tasks of GNN etc? How is n selected?
2) What are the typical graph sizes or the typical sizes for GPU kernels (high-degree
(HD) kernel and low-degree (LD) kernel)? For example, should we expect that 134,103,040 nodes and 268,140,544 edges represent a lower bound or upper bound?
3) While (a), (b), and (c) labels are missing from Figure 7, can the authors comment on why the accuracy is higher for the case of training on 64bit when compared to the 8bit? It would probably make sense to show similar range for both plot on the left and middle. Please also note that I think that the caption text “Figure 7: FPGA mapped dataset results showing (a) memory utilization and (b) accuracy as a function of the number of partitions for CSA multipliers, following the application of FPGA mapping, with a batch size of 1. All the multipliers were trained using 8-bits.” Does not match the plots or should we read the plots from right to left instead of left to right? I apologize to the authors if miss something, but I am confused when reading the caption and trying to understand the plots.
4) One important aspect that the authors could highlight is that whether there is a dependence between the number of partitions and specific features of the graphs like graph size etc in Figure 6. Simply stated, if I am give a graph with 10M nodes and 50M edges, how many partitions should I consider? What other graph features should be considered to solve this problem in the most efficient way?
5) How does the specific motifs in the EDA graph influence the accuracy of the partitions and the overall accuracy of the GNN framework?
6) I apologize to the authors because they have put a lot of effort in writing this paper, but I had a hard time to read the figures (too small text), follow the text explanations and understand some of these sentences: “We utilize a graph partitioning algorithm based on the observation that approximately only 10% boundary edges (nodes) between cluster, to divide the large graphs into smaller sub-graphs for fast GPU processing” what is the observation telling us exactly? Another statement “partitioning does not much impact the accuracy” or “post the 16-partition mark…”. Again, it is probably entirely my fault, but I would recommend a carefully proof reading with a critical eye. Please consider increasing the font sizes of the text in Figures. Please add the identifiers (a) …for Figure 9 to match the caption text “Figure 9: Different multipliers verification time comparisons: (a) CSA Multiplier, (b) Booth Multiplier, (c) 7nm technology mapped multiplier.”

---

### Official Review · Reviewer_8UM8 · 2024-11-05

**Soundness:** 2
**Presentation:** 1
**Contribution:** 2
**Rating:** 3
**Confidence:** 5

**Summary:**

This work propose a joint framework, GROOT, which considers chip design domain knowledge, graph theory, and gpu kernel designs to improve verification efficiency. Compared to prior GNN frameworks that target generic GNN tasks, GROOT with additional circuit knowledge allows optimized partitioning and node feature design achieves much higher performance with high accuracy.

**Strengths:**

-This work presents a degree-based graph partitioning algorithm to split high-degree nodes and low-degree nodes for more efficient GPU optimization respectively.

-This work presents very impressive performance speedup over baseline implementions.

**Weaknesses:**

-This authors argue that the proposed framework has domain specific knowledge included into the GNN optimization. However, the proposed circuit specific optimization including degree-based graph partitioning and node type classification are not new and they are already well studied in prior graph processing and GNN modeling. In addition, these optimizations are mostly specific to the graph structures  and it is not quite relevant to the underlying EDA tasks. Although these approaches do enhance the GNN performance, it is also applicable to generic GNN tasks as long as the graph has varied vertex degree distribution, which is also quite common in social network-based graphs. Hence, the novelty of the proposed framework is limited.

-This work seems to mix various EDA tasks throughout this paper. For instance,  the title indicates logic synthesis, The abstract talks about verification, Then, the experiments mentions multiplier accuracy. Although I know GNNs are intensively explored for circuit representation, I am still confused how GNNs are utilized in these different EDA tasks. More background knowledge is expected before going through the technical details especially for the AI-oriented conference.

-The experiments are all about multipliers. Although data width affects the structure of the circuits substantially, they still share many common blocks. Using small circuit blocks for training and testing on larger circuits with a large number of similar blocks are not quite convincing.



-

**Questions:**

See the weaknesses.

---

### Official Review · Reviewer_tFtM · 2024-11-06

**Soundness:** 2
**Presentation:** 2
**Contribution:** 2
**Rating:** 3
**Confidence:** 4

**Summary:**

The paper presents GROOT, a graph edge re-growth and partitioning framework designed to enhance verification efficiency in large-scale chip design by leveraging graph neural networks (GNNs). GROOT addresses the computational and memory challenges in traditional verification methods by combining domain-specific knowledge of electronic design automation (EDA) with optimized single-GPU processing. The framework includes redesigned node embeddings that incorporate circuit-specific features and uses a tailored partitioning strategy to break down large graphs into manageable sub-graphs, which are then processed using custom GPU kernels optimized for nodes with high and low degrees. Tested on various multipliers, including Carry Save Adder (CSA) and Booth multipliers, GROOT demonstrates significant memory savings and runtime improvements compared to existing approaches like GAMORA and ABC, while maintaining high accuracy levels even for extremely large designs.

**Strengths:**

The paper tries to address a relevant problem and is well written, and well organized.

**Weaknesses:**

1)	The reliance on partitioning and boundary edge re-growth introduces a trade-off between memory efficiency and accuracy. Specifically, as the number of partitions increases, accuracy tends to drop, particularly for complex graphs like Booth multipliers. This suggests that the edge re-growth algorithm may struggle to fully restore the lost connectivity and feature flow between partitions, which could lead to verification errors or degraded GNN performance in highly partitioned graphs.
2)	GROOT’s custom kernel design is tailored for EDA graphs with extreme degree distributions (e.g., nodes with degrees above 512 or below 12). This specialization may reduce efficiency for graphs with less polarized or dynamically varying degree distributions. Additionally, the CUDA implementation with static workload partitioning and tree-based accumulation is optimized for certain degree profiles, which may not generalize well across diverse circuit designs or varying network topologies in EDA applications.
3)	Despite achieving memory reduction via partitioning, the GPU memory requirements for larger circuits (e.g., 1024-bit multipliers) remain substantial. In some cases, even the highest-end NVIDIA A100 GPU (80 GB) approaches its capacity, especially when batch sizes are high. This indicates that GROOT might struggle to handle even larger or more complex circuits without requiring further optimizations or multi-GPU configurations, undermining the claim of single-GPU suitability
4)	GROOT uses the GraphSAGE framework with a fixed set of node features based on the circuit's topology and polarity of input edges. This static approach may limit the adaptability of GROOT's GNN to dynamically changing circuits or circuits with less clear-cut node types (e.g., mixed or non-Boolean gate nodes). Such limitations could hinder GROOT's generalizability to new or unconventional circuit designs beyond those used in the study.
5)	While the paper shows a qualitative trend of accuracy decline with increased partitioning, there’s limited quantitative analysis or formal model on how partition size or graph structure impacts accuracy and memory usage. This makes it challenging to predict how GROOT will perform on circuits with varying topologies, especially if accuracy requirements are stringent.

**Questions:**

1) How does the accuracy of GROOT vary with different levels of partitioning for large graphs, and are there specific partitioning thresholds where accuracy loss becomes significant?

2) How would GROOT's custom GPU kernels perform on circuits with more varied or less extreme degree distributions? Is the framework adaptable to graphs that do not exhibit the same degree polarization?

3) Given the substantial memory requirements for large multipliers, how would GROOT handle larger industrial-scale circuits or multipliers beyond 1024 bits on a single GPU? Would multi-GPU configurations be required?

4) Is there a quantitative model within GROOT to predict the trade-offs between partition size, memory usage, and accuracy for different circuit designs? How might this aid in optimizing GROOT’s performance on unknown circuits?

---

### Note · Authors · 2024-11-15

I have read and agree with the venue's withdrawal policy on behalf of myself and my co-authors.